# Instructor-Blinded Study of Pharmacy Student Learning When a Flipped Online Classroom Was Implemented during the COVID-19 Pandemic

**DOI:** 10.3390/pharmacy10030053

**Published:** 2022-05-11

**Authors:** Paul R. V. Malik, Nardine Nakhla

**Affiliations:** School of Pharmacy, University of Waterloo, 10A Victoria St. S, Kitchener, ON N2G 1C5, Canada; prvmalik@uwaterloo.ca

**Keywords:** learning, flipped classroom, pandemic, pharmacy education

## Abstract

A multi-cohort instructor-blinded research study was completed at the School of Pharmacy, University of Waterloo, to test the impact on study learning endpoints when an online flipped classroom teaching style was implemented during the COVID-19 pandemic. The learning endpoints were gain in factual knowledge and gain in self-confidence in clinical skills (assessing a patient, developing a care plan for a minor ailment, and implementing the care plan by counselling patients on the condition). Gain in factual knowledge was assessed with an instructor-blinded multiple-choice test administered before and after the course. Gain in self-confidence in clinical skills was assessed with a survey asking students to report their self-confidence in completing 10 clinical tasks on a 5-item Likert scale. Students being taught in an online flipped classroom cohort during the COVID-19 pandemic trended toward having a higher gain in self-confidence throughout the course but a lower gain in factual knowledge when compared with a traditional classroom cohort in the previous year.

## 1. Introduction

The COVID-19 pandemic has presented a challenge to both students and instructors in Canadian higher education for over two years now. When the pandemic first hit in the middle of March 2020 and universities were closed across the country, instructors scrambled to move course content into an online format. Most quickly discovered that effective teaching online was not achieved by simply converting the lecture content into video format or narrated slides. Rather, a translational approach was needed to ensure that student learning goals were achieved—one that could perhaps employ a completely different set of instructional tools [1].

Blended learning formats—or ‘flipped’ classrooms—have experienced a surge in popularity as instructors search for ways to maintain effective teaching online. In the context of pharmacy education, flipped classrooms move most of the lecture content to succinct videos while designating in-class time for discussion, problem-solving, and simulated patient cases [2]. In this format, the learner enjoys some degree of autonomy in deciding the pace of learning, but also carries more independent responsibility for learning. In virtually all studies of flipped classrooms in pharmacy education, the final exam grade or subjective measures of perceived learning have been used to measure learning endpoints against historical controls [3]. Under this framework, the flipped classroom teaching style is thought to confer a modest benefit to knowledge-based learning endpoints over the traditional didactic format [3]. In controlled OSCE assessments, the flipped classroom model has further shown benefit to helping students master pharmacy skills, such as clinical pain management or pharmaceutical calculations [4,5].

It is presently unknown what impact the combination of (1) the pandemic and (2) the introduction of online and blended learning formats has had on student learning in pharmacy education. Comparing performance on assignments or exams as a marker of learning when a different teaching style is used is inherently flawed because the investigator—who is also the instructor and interventionalist—writes and scores the assessments themselves. Even in the case of a standardized assessment between two classrooms or teaching styles, the instructor is often unblinded to the content of the assessment and could be biased to emphasize specific course content to one classroom over another that would aid the performance of that cohort and compromise the study.

Therefore, this research aimed to uncover the impact on student learning endpoints when a flipped classroom teaching style was implemented in an advanced patient self-care course during the pandemic. Assessments included an instructor-blinded knowledge test and a qualitative assessment of students’ self-reported confidence in relevant clinical skills administered before and at the conclusion of the course. Fortunately, a control dataset for the course was collected just prior to the pandemic as part of continuous efforts to improve the quality of teaching, and proves a valuable comparator. Under this study design, it is important to note that the specific effects of the two varying factors between cohorts (i.e., the switch to a flipped classroom teaching style and the external impact of the pandemic) cannot be identified separately from one another.

## 2. Materials and Methods

### 2.1. Course Content and Learning Endpoints

The course entitled ‘Advanced Patient Self-Care’ (PHARM 362) is an elective course offered to third-year pharmacy (PharmD) students at the School of Pharmacy, University of Waterloo in Ontario, Canada. This course runs for 12 weeks every winter term, from January until April.

Advanced Patient Self-Care teaches students to critically evaluate the use of prescription (Schedule I) products as well as self-care therapeutic options (e.g., Schedule II and III drugs, unscheduled products, and natural health products) in various disease states and patient populations, with a special focus on minor ailments. Emphasis is placed on the role of the pharmacist in accurately triaging patients in the pharmacy, determining the appropriate use of pharmacologic agents and non-pharmacologic measures, and engaging patients with effective counselling. Considering the evolving pharmacy landscape, the course describes and contrasts the pharmacist-led minor ailment prescribing schemes across Canada. Topics and ailments covered range across dermatology, infectious diseases, gastrointestinal disorders, allergic syndromes, musculoskeletal conditions, nutrition, eye care, and pain management.

The course syllabi for 2020 and 2021 along with specific learning outcomes for the course are included in Appendix A, respectively.

Distilling these learning outcomes down for the study, the key **Learning Endpoints** were:Knowledge: Patient self-care and minor ailment topics;Clinical skill: Patient assessment;Clinical skill: Developing care plans for minor ailments;Clinical skill: Patient counselling for minor ailments.

### 2.2. Study Participants

Third-year pharmacy (PharmD) students were eligible to enroll in the elective course. There were no other prerequisites for the course. In 2020, 55 students were enrolled in the course. Similarly in 2021, 55 students were enrolled in the course.

The study was reviewed and received ethics clearance through a University of Waterloo Research Ethics Committee (ORE#41753) before any subjects were enrolled to participate in the study. Participation in the study was voluntary and was requested through an agent who was not an investigator of the study nor affiliated with the course or its grading. All students completing the assessments submitted informed consent to participate in the study and acknowledged that their scores and participation in the study would not affect their grade in the course. In 2020, 33 students completed the pre-course survey, and 27 students completed the post-course survey. In 2021, 18 students completed the pre-course survey, and 19 students completed the post-course survey.

### 2.3. Didactic Teaching (Control)

In the control cohort in January 2020, course content was delivered over 9 weeks of didactic lectures with 3 weeks of patient case simulations. Content was grouped by ailment and taught as independent modules. Lecture periods were 3 h per week and featured one to three relevant topics of a module. Lectures were delivered in a lecture hall seating the 55 students at the School of Pharmacy, University of Waterloo. Laptops and electronic devices were permitted for notetaking, but recording of the lectures was not permitted. Lecture notes and slides were distributed to students in the online course repository.

In patient case simulations, each student interacted with a standardized patient who was hired to act out a case that required the student to assess the patient, develop a care plan for a minor ailment, and then perform counselling. Students were permitted to use any resources (electronic, text) during this interaction. The student then received feedback from the instructor and from up to six peers who observed the interaction.

Overall course assessments in 2020 were the midterm exam (30% of total grade), final exam (35%), patient case simulations (15%), and individual infographic assignment (Spotlight on Self Care, 20%) [6]. In the 2020 cohort, the final exam was an open-book, non-secure, remote assessment conducted using the Examsoft platform.

In March 2020, all teaching was shifted online at the University of Waterloo. By this point, all lecture content in PHARM 362 had already finished because it occurred over the first 9 weeks. The remaining 3 weeks had previously been allotted for patient case simulations, which proceeded online via Adobe Connect.

### 2.4. Translation to a Flipped Classroom (Intervention)

A flipped classroom model was implemented for online course delivery in 2021.

The course content was organized into six themed modules. Each module began with an independent e-learning component that students were required to review prior to the weekly synchronous class time. The e-learning modules were templated for consistency and contained the following headings: title, learning outcomes, background information, content lessons, activities and assignments, patient resources, and suggested readings and resources. The content lessons were mainly presented in a text format with relevant images, graphics, and videos presented alongside the text to further elaborate on key concepts. In addition, the content featured emphasis boxes with icons for key messages such as ‘Therapeutic tip’, ‘Did you know?’, ‘Clinical pearl’, and ‘Check your knowledge’. The ‘Check your knowledge’ boxes contained non-graded multiple-choice questions that served as formative assessments. Summaries of care plans were presented in the form of process diagrams (e.g., an algorithm for the management of tick bites). Each image, graphic, video, or process diagram was supported by captioning to ensure student accessibility.

The weekly synchronous online class time (3 h per week) was dedicated to the application of knowledge in case studies, facilitated group discussions, and question and answer periods. Care plans for the case studies were formulated collaboratively with the students and included a comprehensive approach to self-care covering schedule I (Rx) drugs, non-prescription drugs, natural health products, and non-pharmacologic strategies for prevention and treatment.

Patient case simulations were conducted online in the same way in 2021 as they were in 2020.

Overall course assessments in 2021 were the midterm exam (30% of total grade), final exam (35%), patient case simulations (15%), and a group literature review and video assignment (Debunking Self Care Myths, 20%). In the 2021 cohort, both the midterm and the final exams were open-book, non-secure, remote assessments conducted using the Examsoft platform.

### 2.5. Qualitative Assessment of Self-Confidence in Clinical Skills Gained

A 10-question survey asked students to report their self-confidence in performing specific tasks that would fulfill **Learning Endpoints 2–4**. A 5-item Likert scale was used to gauge self-confidence (1 = not at all confident, 2 = slightly confident, 3 = moderately confident, 4 = very confident, 5 = extremely confident). With a leading prompt of ‘*How confident are you in your ability to…*’, the 10 tasks evaluated were:Collect a comprehensive and accurate history (medical, medication) by interviewing a patient;Assess and triage a patient’s chief complaint/presenting illness;Develop a treatment plan for a minor ailment involving non-prescription drugs;Develop a treatment plan for a minor ailment involving natural health products;Develop a treatment plan for a minor ailment involving prescription drugs (Schedule I);Develop a treatment plan for a minor ailment involving non-pharmacologic measures;Counsel patients regarding the use of non-prescription drugs to treat a minor ailment;Counsel patients regarding the use of natural health products to treat a minor ailment;Counsel patients regarding the use of prescription drugs (Schedule I) to treat a minor ailment;Counsel patients regarding the use of non-pharmacologic measures to treat a minor ailment.

Students in the 2020 cohort (control) and 2021 cohort (intervention) who agreed to participate in the study were asked to complete the Likert scale assessment before and after the course. The Likert scale assessment was administered as a pre-course survey on Qualtrics in week 1 of the course, and again as a post-course survey in the week leading up to the final exam. Self-confidence in clinical skills gained by learning in the course was measured by the difference in the post-course and pre-course Likert scores. Each student’s entry was coded with a unique four-digit anonymous identifier so that post-course scores could be matched with the relevant pre-course scores in the following within-subject analysis.

At the end of the study, incomplete student surveys were discarded.

Relative difference in self-confidence in clinical skills gained between the 2021 and 2020 cohorts was calculated by:(Gain_2021_ − Gain_2020_)/Gain_2020_

### 2.6. Quantitative Assessment of Knowledge Gained

An instructor-blinded multiple-choice test of 20 questions was created by the teaching assistant for the course, who was also a community pharmacist and had access to all course materials posted on the learning management system. The 20 questions were designed to test the factual knowledge of patient self-care and minor ailment topics (**Learning Endpoint 1**). The instructor was not permitted to have any knowledge of the 20 questions. The instructor was blinded from the assessment from the beginning until the end of the study (a period lasting over two years). The instructor had no role in creating the questions and did not view them until the full statistical analysis of the data was complete. This blinding avoids instructor bias in teaching that could lead students in one cohort to perform well on the specific content of the 20 questions. The director of curriculum on the faculty reviewed the questions for clinical accuracy and relevance to the core concepts of the course. The role of the curriculum director was to verify (1) the clinical accuracy of the questions, (2) that the questions covered the key course content and (3) that the questions were of a suitable level of difficulty for students in the 3rd year of the PharmD program. Questions were one-dimensional in nature, with no problem solving or critical thinking elements. The questions tested factual knowledge evenly from all 12 weeks of course content. The test is presented in Appendix A.

Students in the 2020 cohort (control) and 2021 cohort (intervention) who agreed to participate in the study were asked to complete the test before and after the course. The test was administered as a pre-course survey on Qualtrics in week 1 of the course, and again as a post-course survey in the week leading up to the final exam. Knowledge gained by learning in the course was measured by the difference in the post-course and pre-course test scores. Each student’s entry was coded with a unique four-digit anonymous identifier so that post-course scores could be matched with the relevant pre-course scores in the following within-subject analysis.

At the end of the study, incomplete student surveys were discarded. Three multiple-choice questions from the factual knowledge assessment were excluded from the analysis. The reasons were:Course content was taught in the 2020 cohort but was not taught in the 2021 cohort (n = 1 question);Conflicting therapeutic information was taught during the course that compromised student performance on the question (n = 1 question);Over 90% of students answered the question correctly in the pre-course assessment, limiting the ability to calculate improvement by teaching and learning (n = 1 question).

Relative difference in knowledge gained between the 2021 and 2020 cohorts was calculated by:(Gain_2021_ − Gain_2020_)/Gain_2020_

### 2.7. Within-Subject Paired Analysis

A statistical analysis of within-subject (paired) data was performed using MATLAB R2020a. The data were cut to include only those participants who completed both the pre-course and post-course surveys in the appropriate year, and the paired gains in knowledge and self-confidence in clinical skills throughout the course were calculated. Plotting histograms as well as the Shapiro–Wilk test were used confirm normality of the paired data in each cohort prior to analysis. A two-sided *t* test with pooled variance was performed to test whether the differences in mean knowledge gained and self-confidence in clinical skills gained were statistically significant between the 2020 and 2021 cohorts. In recognition that the Likert scores are not continuous but rather ordinal, a Wilcoxon rank sum test (equivalent to a Mann–Whitney U test) was also conducted to test whether the difference in median self-confidence in clinical skills gained was statistically significant between the 2020 and 2021 cohorts [7].

Finally, the percentages of responders to teaching and learning in each cohort were calculated and presented for each clinical task. Response in each clinical task was defined as achieving a nonzero increase in reported self-confidence at the post-course assessment.

## 3. Results

### 3.1. Qualitative Assessment of Self-Confidence in Clinical Skills Gained

Pre-course (baseline) self-confidence scores were comparable between the 2020 and 2021 cohorts in all ten tasks (Table 1). Post-course self-confidence in clinical skills was higher in the 2021 cohort when compared with the 2020 cohort in 8 out of the 10 tasks surveyed.

Correcting for the pre-course (baseline) scores, the self-confidence in clinical skills gained through teaching and learning in the course was higher in the 2021 cohort than in the 2020 cohort for all 10 tasks surveyed. From the beginning to the end of the course, students in the 2021 cohort reported an average gain in self-confidence of 1.05 points per task on the 5-item Likert scale, whereas students in the 2020 cohort reported an average gain in self-confidence of 0.67 points per task. The mean relative difference in self-confidence gained was +74%. In other words, after participating in the course, the students’ gain in self-confidence in assessing, treating, and counselling on minor ailments was 74% higher during the pandemic in an online flipped classroom setting than would otherwise be achieved through didactic lecturing.

The result suggests that teaching and learning were more effective for fulfilling **Learning Endpoints 2–4** in the flipped classroom and during the pandemic than teaching and learning in a didactic format the year prior.

### 3.2. Quantitative Assessment of Knowledge Gained

The pre-course (baseline) factual knowledge was modestly higher in the 2021 cohort than in the 2020 cohort (56.9% vs. 49.2% average score on the 17 instructor-blinded multiple-choice test) (Table 1). However, the 2021 cohort did not sustain a higher mean score than the 2020 cohort by the end of the course. Post-course factual knowledge was lower in the 2021 cohort when compared with the 2020 cohort (70.6% vs. 76.7% average score on the 17 instructor-blinded multiple-choice test).

Correcting for the pre-course (baseline) scores, the knowledge gained through teaching and learning in the course was considerably lower in the 2021 cohort than in the 2020 cohort. From the beginning to the end of the course, students in 2021 improved scores from 56.9% to 70.6% on the instructor-blinded test, whereas students in the 2020 cohort improved scores from 49.2% to 76.7%. The mean relative difference in knowledge gained was −50%. In other words, the value that students gained from the course towards increasing knowledge on patient self-care and minor ailment topics was 50% lower during the pandemic in an online flipped classroom setting than would otherwise be achieved through didactic lecturing.

The result suggests that teaching and learning were less effective for fulfilling **Learning Endpoint 1** in the flipped classroom and during the pandemic than teaching and learning in a didactic format the year prior.

### 3.3. Within-Subject (Paired) Analysis

Reviewing the within-subject (paired) scores from before and after the course did not change the interpretation of the results (Table 2). Students in 2021 had lower knowledge gained than students in 2020, and students in 2021 had higher self-confidence gained in 8 out of the 10 tasks than students in 2020.

At an alpha level of 0.05, the observations did not achieve statistical significance. This result is most likely driven by the low number of students in 2021 who completed both the pre- and post-course surveys (n = 9). At an alpha level of 0.10, students in 2021 had a significantly higher gain in self-confidence toward developing a treatment plan for a minor ailment involving non-prescription drugs and counselling patients regarding the use of natural health products to treat a minor ailment.

Figure 1 presents the mean and distributions of the changes in scores achieved on the instructor-knowledge test between the 2020 and 2021 cohorts of the study, highlighting that students in 2020 gained more factual knowledge in the course than did students in 2021.

With response to teaching and learning defined by any nonzero increase in reported self-confidence on the post-course Likert assessment, a percentage of ‘responders’ was calculated for each clinical task (Table 3). For 8 of the 10 tasks where students were asked to declare self-confidence, there was a greater proportion of responders to the teaching and learning in the 2021 cohort (intervention) when compared with the 2020 cohort (control). The average response rate to teaching and learning per task was 79% in 2021, compared with 66% in 2020.

## 4. Discussion

In this study, scientifically robust methods were applied pre- and post-course to compare learning endpoints for pharmacy students who were taught in an online flipped classroom format during the COVID-19 pandemic against the learning endpoints of students who were taught in person and in didactic format the year prior. The analysis was performed for a third-year elective course on advanced patient self-care (PHARM 362) at the School of Pharmacy, University of Waterloo in Ontario, Canada.

The investigators of the current study are mindful that the study as designed cannot uniquely identify the effects of switching to an online flipped classroom from the effects of the pandemic on student learning. Instead, the results capture the cumulative effects of both factors on pharmacy student learning endpoints. For example, low knowledge gained may be due to the poor self-study habits of students in the online flipped classroom format, or it could be associated with the impact of a stressful global pandemic on student study habits.

At the conclusion of the course, scores on the instructor-blinded test for factual knowledge (**Learning Endpoint 1**) tended to be lower for students taught during the pandemic (70.6% vs. 76.7%, N.S.). Correcting for pre-course (baseline) scores, students taught during the pandemic had 50% lower knowledge gained throughout the course (students’ scores increased by 13.7 percentage points in 2021 vs. 27.5 percentage points in 2020). The within-subject (paired) analysis could not confirm statistical significance of the trend in knowledge gained due to the low number of students who completed both the pre-course and post-course surveys in the intervention cohort.

On the other hand, student scores for self-confidence in clinical skills (**Learning Endpoints 2–4**) as assessed using a 5-item Likert scale were higher for students taught in the online flipped classroom format during the pandemic (N.S.). Correcting for pre-course Likert scores, students in the 2021 cohort had a mean 74% higher self-confidence gained throughout the course (mean increase in Likert score per task was 1.05 points in 2021 vs. 0.67 points in 2020). Similar to the knowledge test, the within-subject (paired) analysis could not confirm statistical significance of the trend. With response to teaching and learning, defined as any nonzero increase in self-confidence in clinical skills by the end of the course, the average rate of response to teaching and learning per task was 79% in 2021 compared with 66% in 2020. Overall, the data show clear trends that students taught in an online flipped classroom format during the pandemic had greater gains in self-confidence through teaching and learning than did those taught in a didactic format the year prior.

However, the higher self-confidence did not necessarily translate to better performance on the patient case simulations or on the final exam. In a retrospective analysis of grades for all students in the course (n = 55 in both 2020 and 2021), mean grades on the patient cases used for the simulations and the final exams were nearly identical between the two cohorts. This result is presented post hoc and the relevant limitations to measuring grades as assigned by the investigator(s) should be acknowledged as discussed before.

Contributing reasons to the poor performance in knowledge gained are not yet known. As mentioned, factors from the pandemic and from the implementation of the online flipped classroom may contribute, as well as multiplicative factors from both sources. The pandemic has affected higher education in more ways than simply forcing instruction to be moved to an online format. The stress of a global pandemic has unknown impacts on student study habits and course engagement. Despite their best efforts, instructors have struggled to maintain engagement, encourage deep learning, and preserve the integrity of assessments during the pandemic. Academic integrity and overall engagement have historically suffered in pre-pandemic remote assessments and in flipped classroom formats, too [2,8]. These problems are only magnified by the pandemic [9]. If students can achieve high grades on exams (>80%) while exhibiting low course engagement, their self-confidence may increase, and deep learning and retention of knowledge may decrease.

The literature describing the quantitative impact of the pandemic on learning in higher education is understandably rare at present, due to difficulty with matching historical controls. In a third-year elective course taught remotely during the pandemic, pharmacy student grades in online OSCEs (patient case simulations) and on the final exam were higher but not significantly different from the grades in the previous cohort that was taught and evaluated in person [10]. This result supports that found in the present study that self-confidence in clinical skills (and thereby perhaps the learning of clinical skills) have not been compromised by the shift to online learning that occurred due to the pandemic.

In the field of economics, standardized assessments at four R1 institutions in the United States revealed that scores of students educated during the pandemic were an average of 0.2 standard deviations below the scores of students from 2019 [11]. This result agrees with our finding that factual knowledge gained through courses during the pandemic may be lower than through courses taught in person before the pandemic, although our finding must be taken in the context of the additional shift from a lecture-based teaching method to a flipped classroom and short-video-based teaching method in which the onus is on the students to complete some of the learnings independently before classes.

As mentioned, a low number of students in the flipped classroom cohort completed both the pre-course and post-course surveys (n = 9), limiting the determination of statistical significance. Low survey completion in the 2021 cohort may reflect poor student engagement with the course as a result of COVID-19 or as a result of participating in a flipped classroom format. This issue warrants future investigation.

This study highlights the opportunity for teaching assistants to be involved in more scientifically rigorous assessments of learning when teaching is impacted by external factors, or when a new teaching style is applied. Instructor-blinded assessments of learning are simple to craft and simple to execute. Of interest, none of the students in both cohorts were able to achieve 100% on the post-course assessment of factual knowledge, despite the questions being focused on core concepts and being one-dimensional in nature.

## 5. Conclusions

Students being taught in an online flipped classroom cohort during the COVID-19 pandemic trended toward having a higher gain in self-confidence throughout the course but a lower gain in factual knowledge when compared with a traditional classroom cohort in the previous year.

Although most higher education institutions are now returning to in-person education, the ripple effects of potential gaps in student knowledge that occurred throughout the past two years may require remediation if consistently observed on a larger scale. Results of standardized assessments—such as licensing exams for pharmacists—may be informative toward this end.

## Figures and Tables

**Figure 1 pharmacy-10-00053-f001:**
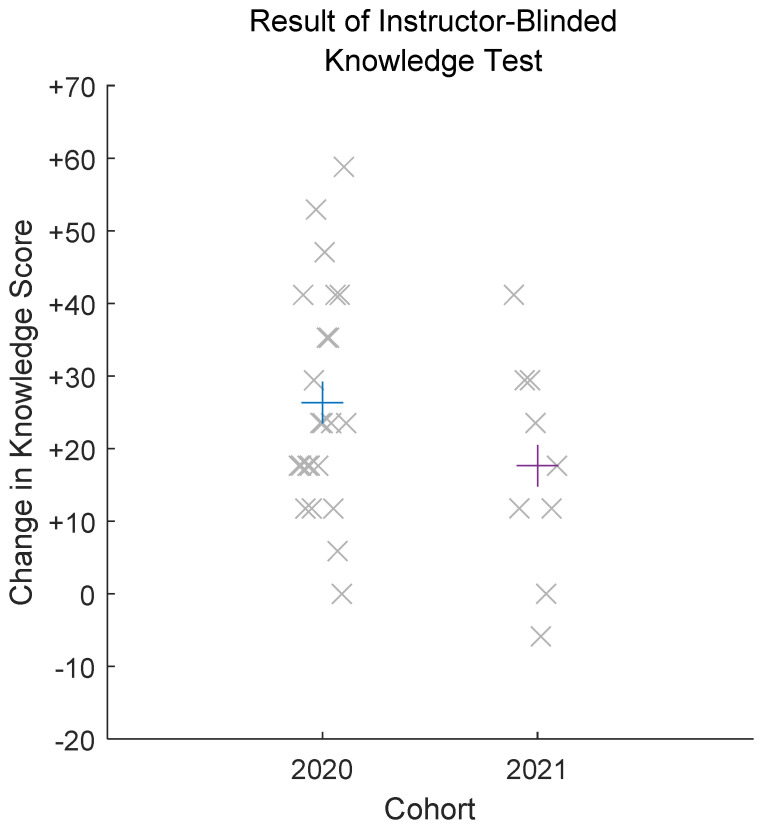
Result of instructor-blinded knowledge test (paired data only) among participants in 2020 and 2021. Observed data are presented as the mean knowledge gained on the instructor-blinded test (plus sign) overlaid with individual data for knowledge gained on the instructor-blinded test (X sign).

**Table 1 pharmacy-10-00053-t001:** Comparison of knowledge gained and self-confidence in clinical skills gained between the control (2020) and intervention (2021) cohorts.

Task	Control Cohort (2020)			Intervention Cohort (2021)			Relative Difference in Gain
	Pre-Course (n = 33)	Post-Course (n = 27)	Gain	Pre-Course (n = 18)	Post-Course (n = 19)	Gain	
Factual Knowledge							
Multiple-choice questions (17) covering course content	49.2% ± 10.7% [29.4% to 70.6%]	76.7% ± 13.1% [52.9% to 100%]	+27.5%	56.9% ± 15.2% [41.2% to 88.2%]	70.6% ± 17.2% [41.2% to 100%]	+13.7%	−50%
Patient Assessment							
Collect a comprehensive and accurate history (medical, medication) by interviewing a patient	3.58 ± 0.56 [3 to 5]	4.0 ± 0.68 [3 to 5]	+0.42	3.56 ± 0.92 [1 to 5]	4.10 ± 0.57 [3 to 5]	+0.54	+29%
Assess and triage a patient’s chief complaint/presenting illness	3.0 ± 0.61 [2 to 4]	3.74 ± 0.76 [2 to 5]	+0.74	2.83 ± 0.79 [1 to 4]	3.63 ± 0.60 [3 to 5]	+0.80	+8%
Developing a Care Plan							
Develop a treatment plan for a minor ailment involving non-prescription drugs	2.94 ± 0.70 [1 to 4]	3.52 ± 0.85 [2 to 5]	+0.58	2.44 ± 0.78 [1 to 3]	3.74 ± 0.73 [2 to 5]	1.30	+124%
Develop a treatment plan for a minor ailment involving natural health products	1.97 ± 0.77 [1 to 4]	2.56 ± 0.85 [1 to 4]	+0.59	1.61 ± 0.70 [1 to 3]	2.37 ± 1.12 [1 to 5]	+0.76	+29%
Develop a treatment plan for a minor ailment involving prescription drugs (Schedule I)	2.33 ± 0.85 [1 to 4]	3.37 ± 0.74 [2 to 5]	+1.04	2.11 ± 0.83 [1 to 3]	3.37 ± 0.76 [2 to 5]	+1.26	+21%
Develop a treatment plan for a minor ailment involving non-pharmacologic measures	3.15 ± 0.94 [1 to 5]	4.0 ± 0.88 [2 to 5]	+0.85	2.72 ± 0.75 [2 to 4]	4.11 ± 0.81 [3 to 5]	+1.39	+64%
Patient Counselling							
Counsel patients regarding the use of non-prescription drugs to treat a minor ailment	3.03 ± 0.68 [2 to 4]	3.63 ± 0.74 [2 to 5]	+0.60	2.83 ± 0.71 [1 to 4]	3.90 ± 0.66 [3 to 5]	+1.07	+78%
Counsel patients regarding the use of natural health products to treat a minor ailment	2.24 ± 0.83 [1 to 4]	2.52 ± 0.85 [1 to 4]	+0.28	1.67 ± 0.69 [1 to 3]	2.79 ± 0.85 [1 to 4]	+1.12	+300%
Counsel patients regarding the use of prescription drugs (Schedule I) to treat a minor ailment	2.76 ± 0.75 [1 to 4]	3.52 ± 0.85 [2 to 5]	+0.76	2.61 ± 0.78 [1 to 4]	3.74 ± 0.81 [2 to 5]	+1.13	+49%
Counsel patients regarding the use of non-pharmacologic measures to treat a minor ailment	3.18 ± 0.64 [2 to 4]	4.04 ± 0.81 [2 to 5]	+0.86	2.94 ± 0.73 [2 to 4]	4.11 ± 0.66 [3 to 5]	+1.17	+36%

Observed data are presented as the mean ± SD [range]. Green shading reflects that students in 2021 had higher achievement of learning endpoints than students in 2020; orange shading reflects students in 2021 had lower achievement of learning endpoints than students in 2020.

**Table 2 pharmacy-10-00053-t002:** Comparison of knowledge gained and self-confidence in clinical skills gained between the control (2020) and intervention (2021) cohorts using paired data only.

Task	Control Cohort (2020)	Intervention Cohort (2021)	Point Difference in Mean Paired Improvement [95% Confidence Interval]	*p*-Value for Significance
	Paired (Within-Subject) Gain n = 23	Paired (Within-Subject) Gain n = 9		
Factual Knowledge				
Multiple choice questions (17) covering course content	+26.3% ± 15.3% [+0% to +52.9%]	+17.7% ± 15.0% [−5.9% to +41.2%]	−8.6% [−20.9% to +3.6%]	0.16 ^a^
Patient Assessment				
Collect a comprehensive and accurate history (medical, medication) by interviewing a patient	+0.49 ± 0.67 [−1 to +2]	+0.44 ± 0.73 [−1 to +1]	−0.05 [−0.58 to +0.51]	0.90 ^a^ 0.96 ^b^
Assess and triage a patient’s chief complaint/presenting illness	+1 ± 0.67 [+0 to +2]	+0.67 ± 0.87 [−1 to +2]	−0.33 [−0.92 to 0.25]	0.26 ^a^ 0.35 ^b^
Developing a Care Plan				
Develop a treatment plan for a minor ailment involving non-prescription drugs	0.74 ± 0.86 [−1 to +3]	+1.33 ± 0.87 [+0 to +3]	+0.59 [−0.10 to +1.29]	0.09 ^a^ 0.08 ^b^
Develop a treatment plan for a minor ailment involving natural health products	0.70 ± 0.82 [−1 to +2]	+0.89 ± 0.78 [+0 to +2]	+0.19 [−0.46 to +0.85]	0.55 ^a^ 0.58 ^b^
Develop a treatment plan for a minor ailment involving prescription drugs (Schedule I)	+1.17 ± 0.83 [+0 to +3]	+1.33 ± 0.71 [+1 to +3]	+0.16 [−0.49 to +0.80]	0.62 ^a^ 0.75 ^b^
Develop a treatment plan for a minor ailment involving non-pharmacologic measures	+0.91 ± 0.90 [+0 to +3]	+1.22 ± 0.67 [+0 to +2]	+0.31 [−0.37 to +0.99]	0.36 ^a^ 0.21 ^b^
Patient Counselling				
Counsel patients regarding the use of non-prescription drugs to treat a minor ailment	+0.65 ± 0.78 [−1 to 2]	+1.11 ± 0.78 [+0 to +2]	+0.46 [−0.17 to +1.08]	0.14 ^a^ 0.17 ^b^
Counsel patients regarding the use of natural health products to treat a minor ailment	+0.39 ± 0.94 [−2 to 2]	+1.0 ± 0.71 [+0 to +2]	+0.61 [−0.10 to +1.32]	0.09 ^a^ 0.11 ^b^
Counsel patients regarding the use of prescription drugs (Schedule I) to treat a minor ailment	+0.87 ± 0.87 [−1 to 3]	+1.11 ± 0.6 [+0 to +2]	+0.24 [−0.41 to +0.89]	0.45 ^a^ 0.38 ^b^
Counsel patients regarding the use of non-pharmacologic measures to treat a minor ailment	+0.96 ± 0.77 [−1 to 2]	+1.11 ± 0.78 [+0 to +2]	+0.15 [−0.47 to +0.77]	0.61 ^a^ 0.66 ^b^

^a^ Calculated by two-sided *t* test. ^b^ Calculated by two-sided Wilcoxon rank sum test. Observed data are presented as the mean ± SD [range]. Green shading reflects that students in 2021 had higher achievement of learning endpoints than students in 2020; orange shading reflects students in 2021 had lower achievement of learning endpoints than students in 2020.

**Table 3 pharmacy-10-00053-t003:** Percent of responders to teaching and learning (paired data only) among participants in 2020 and 2021.

Task	Control Cohort (2020)	Intervention Cohort (2021)
	Response Rate (%) n = 23	Response Rate (%) n = 9
Patient Assessment		
Collect a comprehensive and accurate history (medical, medication) by interviewing a patient	11/23 (48%)	5/9 (56%)
Assess and triage a patient’s chief complaint/presenting illness	18/23 (78%)	6/9 (67%)
Developing a Care Plan		
Develop a treatment plan for a minor ailment involving non-prescription drugs	14/23 (61%)	8/9 (89%)
Develop a treatment plan for a minor ailment involving natural health products	13/23 (57%)	6/9 (67%)
Develop a treatment plan for a minor ailment involving prescription drugs (Schedule I)	18/23 (78%)	9/9 (100%)
Develop a treatment plan for a minor ailment involving non-pharmacologic measures	15/23 (65%)	8/9 (89%)
Patient Counselling		
Counsel patients regarding the use of non-prescription drugs to treat a minor ailment	15/23 (65%)	7/9 (78%)
Counsel patients regarding the use of natural health products to treat a minor ailment	13/23 (57%)	7/9 (78%)
Counsel patients regarding the use of prescription drugs (Schedule I) to treat a minor ailment	16/23 (70%)	8/9 (89%)
Counsel patients regarding the use of non-pharmacologic measures to treat a minor ailment	18/23 (78%)	7/9 (78%)
Average		
Average response rate per task	15.1/23 (66%)	7.1/9 (79%)

Green shading reflects that students in 2021 had higher achievement of learning endpoints than students in 2020; orange shading reflects students in 2021 had lower achievement of learning endpoints than students in 2020.

## Data Availability

Data from this study are not available for sharing due to ethical, legal, and privacy restrictions.

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
