# Peer review of "Instructor-Blinded Study of Pharmacy Student Learning When a Flipped Online Classroom Was Implemented during the COVID-19 Pandemic"

_pharmacy, 2022, doi:10.3390/pharmacy10030053_

Round 1
Reviewer 1 Report
The research topic of this study is extremely important, as although the pandemic might be soon over, there is no going back with teaching methods. Although the planning and executing this study is of high quality, two issues are bothering me here. Could you elaborate more and possibly compare the difference between traditional lecturing and flipped classroom method in prepandemic circumstances? You could add some literature on this issue.
Second concern is related to the first one. Now I see two profound confounders here, namely the change of teaching method from traditional to flipped classroom and the pandemic conditions at the same time. So, as there seems to be less committed students in 2021, judging by the survey outcomes - is this related to the pandemic or to the change of the teaching method - or possibly due to both changes?
To me, the level of commitment among the students in 2021 class, seems to confirm also our observations: some students excel better when they can work on the own in remote circumstances, but the percentage of these students is low. The worry is for the bigger mass of students who seem to disappear and who cannot be easily reached during the remote learning.
Reviewer 2 Report
Interesting topic. Some sections lacking clarity making if difficult for readers to have a clear understanding of your work e.g.,
- line 77 - these are not learning outcomes (which must be student-focused, measurable and achievable) please re-write. The "mapping" of the study questions to the so-called learning outcomes wasn't clearly explained.
- line 181 - did the instructor review and/or approve the MCQ? It seems they were blinded to the student responses, but does instructor-blinded mean they had no input? Please explain.
- line 188 - what was evenly distributed over the 12 weeks?
- line 285 - reword "strongest learning outcomes" to highest achievement of learning outcomes or similar wording
- discussion - line 11 - is this "-50% lower knowledge gained"? Similar expression in line 19.
Re: methodology - line 185 - is review by a single individual (curriculum director) considered appropriate validation? Any reference for this?
Re: results - line 243 - I don't believe you can generalize - only describe the results actually seen.
Discussion - suggest adding more information about impact of the pandemic on student performance, level of engagement etc. Make it clearer why readers should pay attention to these results (re: future use of flipped classroom models etc).
consider adding recommendations for other instructors based on these results and pandemic experiences.
Reviewer 3 Report
This study is well executed and presented. It is difficult to determine subjective views of students especially during a pandemic situation and you have managed this successfully. My concern is the relatively small sample size and robustness of statistical analysis. However it is a well written and well- thought out study. It provides literature on difficulties of learning and teaching during major global events and perhaps how best adapt or overcome these situations.
Author Response
Reviewer comment: "This study is well executed and presented. It is difficult to determine subjective views of students especially during a pandemic situation and you have managed this successfully. My concern is the relatively small sample size and robustness of statistical analysis. However it is a well written and well- thought out study. It provides literature on difficulties of learning and teaching during major global events and perhaps how best adapt or overcome these situations."
Dear reviewer,
Thank you for taking the time to provide insight on the manuscript and the study. We agree that the small sample size is limiting to the statistical conclusions. Statistical significance was not achieved, and we are mindful to state that the findings are just trends. The trends are in fact consistent across the specific endpoints (e.g., there are consistent trends across the 10 tasks assessed for self-confidence in clinical skills). Interestingly, we do note that the sample size of students who fully participated in the study in the second cohort may be a finding on its own that could be representative of the level of student engagement with the course in the intervention cohort.
Regarding the statistical methodology, we did receive a statistical consultation on the selected methods for each endpoint (factual knowledge scores, Likert scale scores). If you have additional considerations for a robust statistical analysis, we can certainly consider it.
Thank you again for the insight and review.